# FUTURE POLICY AWARE PREFERENCE LEARNING FOR MATHEMATICAL REASONING

## ABSTRACT

Preference learning methods such as Direct Preference Optimization (DPO) have become standard for Large Language Model (LLM) post-training, yet they are often ineffective for mathematical reasoning. A key challenge is the large token overlap between preferred and dispreferred trajectories; lowering the probability of dispreferred trajectories also reduces the probability of shared useful tokens, leading to over-penalization and overall performance collapse. As a mitigation, existing algorithms include the probability of a trajectory under the current policy as a regularization term, which decreases the effect of the gradient when the probability is low. However, by the time this effect takes hold, useful tokens may have already been over-penalized as the model has begun to degrade. To address this, we propose **Future Policy Aware (FPA)** preference learning, which replaces the current policy with a future policy in the regularization term. This future policy is estimated via lightweight, logit-space extrapolation from a reference model toward the current model. FPA enables safer training by preemptively regularizing potentially problematic gradients. We apply FPA to DPO, RPO, and SimPER and evaluate them on the MATH and GSM8K benchmarks. FPA yields consistent performance gains, with the largest improvements observed with SimPER, achieving gains of up to 5.75%. We demonstrate that FPA provides proactive regularization while preserving the probability of shared, useful mathematical tokens, and enables longer, degradation-free training with negligible computational overhead. We will release our code publicly upon publication.

## 1 INTRODUCTION

Preference learning methods such as Direct Preference Optimization (DPO) (Rafailov et al., 2023) have become a standard for LLM post-training, with success across various domains like instruction-following, summarization, and model safety (Tunstall et al., 2023; Lambert et al., 2024). However, they have been relatively ineffective for mathematical reasoning—often even degrading the model's capabilities (Lai et al., 2024). One of the main problems is *gradient entanglement* (Yuan et al., 2025): preferred and dispreferred mathematical reasoning trajectories share substantial tokens (e.g., equations, intermediate steps, symbols), so decreasing the probability of a dispreferred trajectory can also decrease the probability of shared tokens in preferred trajectories (Xu et al., 2024; Pal et al., 2024).

When preferred and dispreferred trajectories share many tokens, their gradients become highly correlated and point in similar directions. As a result, the update directions for these gradients become nearly opposite, as the objective encourages the preferred while discouraging the dispreferred (see Figure 1). This large angle between the preferred and dispreferred gradients is the source of training instability under gradient entanglement, as it makes the final update direction—effectively their vector sum—highly sensitive to their relative magnitudes. Consequently, the larger gradient—often from dispreferred trajectories—can dominate the update and cause performance degradation. Our empirical analysis on mathematical datasets confirms this (see Figure 2). We find a large angle between the gradients, and the gradient norm of the dispreferred trajectory is larger.

Most preference learning algorithms already include mechanisms that regulate gradient magnitudes. For instance, DPO (Rafailov et al., 2023) applies a sigmoid to a reference-normalized log-ratio that regulates the update magnitude (step size) as the margin increases, and SimPER (Xiao et al., 2025)

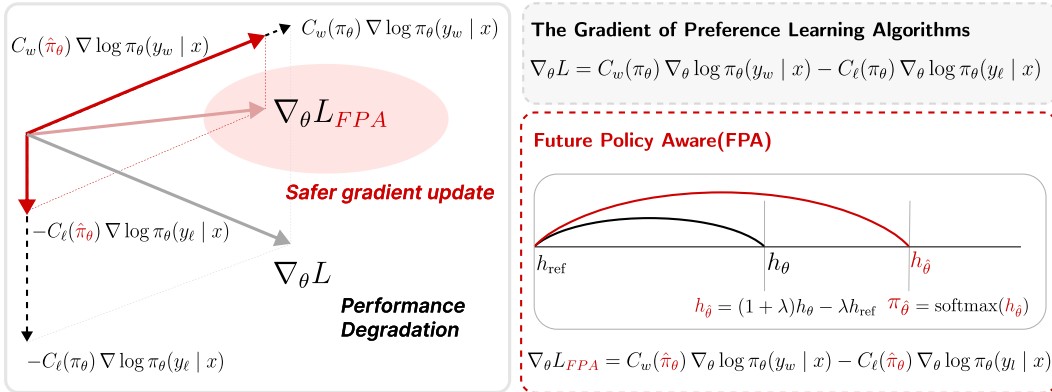

Figure 1: (Left) Under gradient entanglement, a large angle between the preferred and dispreferred gradients makes the update angle more sensitive to their relative magnitudes. (Right) FPA mitigates this instability by constraining the relative magnitudes. $C(\pi_\theta)$ denotes the algorithm specific coefficient (see § 2), and $y_w, y_l$ denote the preferred and dispreferred trajectories respectively.

uses inverse perplexity coefficients to regulate dispreferred updates (see Table 1). However, these regularizations are often insufficient in mathematical domains. To mitigate this problem, some prior approaches focused on strengthening preferred gradients in particular. For example, Reasoning Preference Optimization (RPO) (Pang et al., 2024) augments the DPO loss with a negative log-likelihood term for preferred samples; DPO-Positive (Pal et al., 2024) adds a term that prevents the preferred gradient from decreasing. Nevertheless, the dispreferred gradient remains only indirectly controlled and susceptible to instability.

Despite such efforts, modifying the policy representation itself for regularization remains under-explored. In many existing preference algorithms, the regularization effect is proportional to the sample's probability calculated by the *current policy*. Thus, it mitigates exploding dispreferred gradients when the probabilities of dispreferred samples are too low, and complements preferred gradients when the probabilities of preferred samples are high (see Table 1). However, this effect is rather reactive; by the time a low probability triggers this mechanism, useful, shared tokens may have already been over-penalized. To address this, we propose **Future Policy Aware** (**FPA**) preference learning, which estimates a future policy via lightweight linear extrapolation from the reference model's logits toward the current model's logits (Liu et al., 2024; Kim et al., 2025) (see Figure 1). This estimation captures where the policy is about to move under the ongoing update. By using this future policy to weight the gradients in the loss, our approach makes regularization **proactive rather than reactive**, suppressing problematic updates while preserving each base algorithm's intended mechanism (e.g., DPO's margin shaping, SimPER's inverse-perplexity scaling).

To evaluate the effectiveness of FPA, we apply it to several preference learning algorithms. Specifically, since underlying regularization mechanisms under gradient entanglement may significantly differ depending on how preferred gradients and dispreferred gradients interact with each other within a regularization term, we choose three preference algorithms where preferred and dispreferred gradients have different dynamics. (1) DPO, as a widely used, generic baseline with symmetric gradient regularization (Rafailov et al., 2023), (2) RPO, which strengthens the positive gradient of DPO with a negative log-likelihood term (Pang et al., 2024), (3) SimPER, which asymmetrically and explicitly controls each gradient's magnitude (Xiao et al., 2025). Additional theoretical analysis on algorithms can be found in Appendix B. On the MATH and GSM8K benchmarks, FPA achieves consistent improvements across all algorithms, with the largest gains observed on Sim-PER—averaging 2.58% and reaching gains of up to 5.75%. Empirically, FPA enables preemptive regularization of gradients, preserving the probabilities of shared mathematical tokens. This prevents early downward drift and stabilizes model training, allowing for longer, degradation-free training. Additionally, results demonstrate that FPA delivers adaptive regularization, highlighting that its effect cannot be replaced by simply lowering the learning rate. Overall, FPA offers a broadly applicable, low-overhead solution to gradient entanglement that integrates easily into existing preference learning algorithms.

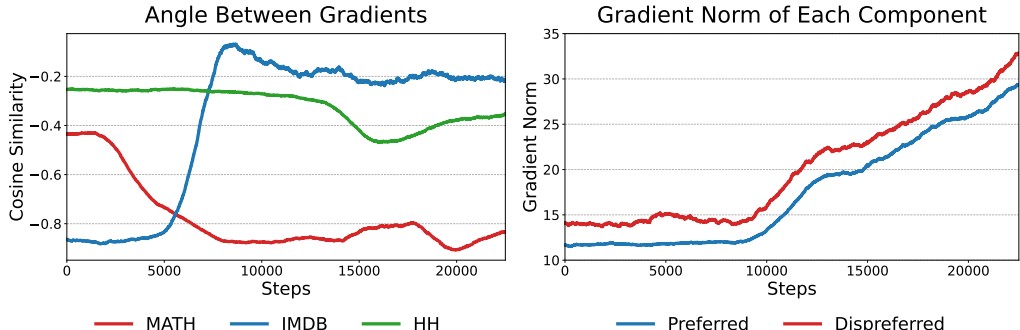

Figure 2: (Left) The angle between preferred gradient and dispreferred gradient is large for the MATH dataset, compared to the Anthropic Helpful and Harmless (HH) dataset and the Stanford IMDb segmentation dataset. (Right) The preferred and dispreferred gradient norms for the MATH dataset. The dispreferred norm is consistently larger. Experimental details are available in Appendix A.

## 2 PRELIMINARIES

**Training Dynamics** The gradient of any preference learning objective can be written in a general form by isolating algorithm-specific terms into coefficients $C_w(\pi_\theta)$ and $C_l(\pi_\theta)$ for the preferred and dispreferred trajectories, respectively (see Table 1):

$$\nabla_\theta \mathcal{L} = -\mathbb{E}_{(x,y_w,y_l)\sim\mathcal{D}} \left[ C_w(\pi_\theta)\nabla_\theta \log \pi_\theta(y_w|x) - C_l(\pi_\theta)\nabla_\theta \log \pi_\theta(y_l|x) \right]. \tag{1}$$

The training dynamics are driven by two factors. (1) $\nabla_\theta \log \pi_\theta(y|x)$ sets the gradient direction and magnitude. Because $\|\nabla_\theta \log \pi_\theta(y|x)\| \propto 1/\pi_\theta(y|x)$, the dispreferred gradient norm can grow as training drives $\pi_\theta(y_l|x)$ down, creating a vicious circle that can explode gradients and cause model collapse (Mao et al., 2025). (2) The coefficient $C(\pi_\theta)$ modulates the gradient magnitudes based on the current policy's probabilities ($\pi_\theta(y|x)$), providing regularization that can prevent overtraining and collapse. Moreover, using different coefficients (i.e., $C_w \neq C_l$) changes the relative magnitudes between preferred and dispreferred gradients, shifting the overall update direction.

**DPO** Direct Preference Optimization (DPO) (Rafailov et al., 2023) simplifies Reinforcement Learning from Human Feedback (RLHF) (Ouyang et al., 2022) into a single loss under a Bradley–Terry preference model (Bradley & Terry, 1952). The loss naturally reduces $C_w = C_l$ as the log-probability margin ($\log \frac{\pi_\theta(y_w|x)}{\pi_{\text{ref}}(y_w|x)} - \log \frac{\pi_\theta(y_l|x)}{\pi_{\text{ref}}(y_l|x)}$) between preferred and dispreferred trajectories grows, thereby regularizing overtraining. The DPO objective is given as follows:

$$\mathcal{L}_{\text{DPO}}(\theta) = -\mathbb{E}_{(x,y_w,y_l)\sim\mathcal{D}} \left[ \log \sigma \left( \beta \log \frac{\pi_\theta(y_w|x)}{\pi_{\text{ref}}(y_w|x)} - \beta \log \frac{\pi_\theta(y_l|x)}{\pi_{\text{ref}}(y_l|x)} \right) \right] \tag{2}$$

**RPO** To mitigate gradient entanglement in reasoning tasks, Reasoning Preference Optimization (RPO) (Pang et al., 2024) strengthens the preferred gradient by augmenting DPO with a negative log-likelihood (NLL) term on $y_w$, weighted by $\alpha$ ($C_w = C_l + \alpha/|y|$). This explicitly discourages decreases in the preferred trajectory's probability. The RPO objective is as follows:

$$\mathcal{L}_{RPO}(\theta) = \mathcal{L}_{DPO}(\theta) - \alpha\mathbb{E}_{(x,y_w,y_l)\sim\mathcal{D}} \left[ \frac{\log \pi_\theta(y_w|x)}{|y_w|} \right] \tag{3}$$

**SimPER** SimPER introduces an inverse-perplexity-based objective that does not require a reference model and additional hyperparameters (Xiao et al., 2025; LG AI Research et al., 2025a;b). Regularization is controlled with the inverse-perplexity term $\text{PER}^{-1}(y|x) = \exp\left(\frac{1}{|y|}\log \pi_\theta(y|x)\right)$ as the $C$. As $\pi_\theta(y_l|x)$ decreases, $\exp\left(\frac{1}{|y_l|}\log \pi_\theta(y_l|x)\right)$ also decreases, regularizing the growth of $\|\nabla_\theta \log \pi_\theta(y_l|x)\|$. The SimPER objective is as follows:

$$\mathcal{L}_{\text{SimPER}}(\theta) = -\mathbb{E}_{(x,y_w,y_l)\sim\mathcal{D}} \left[ \exp\left(\frac{1}{|y_w|}\log \pi_\theta(y_w|x)\right) - \exp\left(\frac{1}{|y_l|}\log \pi_\theta(y_l|x)\right) \right]. \tag{4}$$

Table 1: Algorithm-specific coefficients $C$ for different preference learning methods. Here $\sigma(\cdot)$ denotes the sigmoid function, $\beta$ is the DPO hyperparameter, and $\alpha$ controls the weight of the negative log-likelihood in RPO. $|y_w|$ and $|y_l|$ denote the lengths of the preferred and dispreferred trajectories.

| Algorithm | Preferred $C_w$ | Dispreferred $C_l$ |
|---|---|---|
| DPO | $\beta\sigma\left(\beta\log\frac{\pi_\theta(y_l\mid x)}{\pi_{\text{ref}}(y_l\mid x)} - \beta\log\frac{\pi_\theta(y_w\mid x)}{\pi_{\text{ref}}(y_w\mid x)}\right)$ | $\beta\sigma\left(\beta\log\frac{\pi_\theta(y_l\mid x)}{\pi_{\text{ref}}(y_l\mid x)} - \beta\log\frac{\pi_\theta(y_w\mid x)}{\pi_{\text{ref}}(y_w\mid x)}\right)$ |
| RPO | $\beta\sigma\left(\beta\log\frac{\pi_\theta(y_l\mid x)}{\pi_{\text{ref}}(y_l\mid x)} - \beta\log\frac{\pi_\theta(y_w\mid x)}{\pi_{\text{ref}}(y_w\mid x)}\right) + \alpha/|y_w|$ | $\beta\sigma\left(\beta\log\frac{\pi_\theta(y_l\mid x)}{\pi_{\text{ref}}(y_l\mid x)} - \beta\log\frac{\pi_\theta(y_w\mid x)}{\pi_{\text{ref}}(y_w\mid x)}\right)$ |
| SimPER | $\frac{1}{|y_w|}\exp\left(\frac{1}{|y_w|}\log\pi_\theta(y_w\mid x)\right)$ | $\frac{1}{|y_l|}\exp\left(\frac{1}{|y_l|}\log\pi_\theta(y_l\mid x)\right)$ |

It is worth highlighting the differences in regularization dynamics across these methods based on their coefficient structure. In DPO, the scaling is applied symmetrically to both gradients ($C_w = C_l$). In RPO, the preferred term is further strengthened by $\alpha$ ($C_w = C_l + \alpha/|y|$). In SimPER, the coefficients are inherently different ($C_w \neq C_l$), enabling independent control of the preferred and dispreferred gradients.

## 3 FUTURE POLICY AWARE PREFERENCE LEARNING

In this section, we introduce **Future Policy Aware (FPA)** preference learning to address gradient entanglement in mathematical reasoning. From Equation 1, the coefficient $C$ controls gradient magnitudes based on the current policy, $\pi_\theta$. However, this makes the regularization reactive—it often engages too late, after shared, useful tokens have already been over-penalized. To remedy this, we propose making the regularization proactive by replacing the current policy $\pi_\theta$ with a predicted future policy $\hat{\pi}_\theta$ when computing the coefficient $C$. We estimate this future policy using a lightweight logit-space extrapolation, following recent works (Liu et al., 2024; Kim et al., 2025):

$$\hat{\pi}_\theta = \text{softmax}\big((1+\lambda)h_\theta - \lambda h_{\text{ref}}\big) \tag{5}$$

where $\hat{\pi}_\theta$ is extrapolated from the reference logits ($h_{\text{ref}}$) towards the current logits ($h_\theta$) with a strength controlled by $\lambda$. This creates an adaptive, look-ahead regularization. By recomputing the coefficient under $\hat{\pi}_\theta$, our method anticipates problematic updates and restrains the corresponding gradient earlier. For example, if the probability of a dispreferred trajectory is currently high (e.g., 0.7) but abruptly drops in a few future updates (e.g., to 0.1), standard regularization only engages after the probability becomes lower. By that point, shared, useful tokens may have been over-penalized. In contrast, FPA anticipates this drop and begins regularization in advance, preventing potential model degradation. Thus, FPA functions as an *early brake*, while preserving the underlying regularization dynamics.

FPA yields primary benefits that vary depending on the structure of the underlying preference learning algorithm. (1) For algorithms like DPO with symmetric coefficients ($C_w = C_l$), FPA acts as an early brake; through the future policy $\hat{\pi}_\theta$, it preemptively reduces the magnitude of the shared coefficient $C$ to ensure a safer overall update step. (2) Conversely, for algorithms like RPO and SimPER with asymmetric coefficients ($C_w \neq C_l$), FPA provides a more targeted regularization. It primarily suppresses the dispreferred coefficient ($C_l$)—often the source of instability—which corrects the update direction itself by controlling the relative magnitude of preferred and dispreferred gradients. While FPA could theoretically help strengthen the preferred trajectory via SimPER's coefficient ($C_w$), this effect is marginal in practice (see § 4.2). Because the update is dominated by the larger gradient norm of the dispreferred trajectory, FPA's primary contribution lies in regularizing the dispreferred gradient.

For practical implementation, we define FPA using the stop-gradient operator. Specifically, we replace $\pi_\theta$ with $\hat{\pi}_\theta$ in the coefficient computation while preserving the main gradient flow through $\pi_\theta$. From Equations 1 and 5, the general form for FPA preference learning can be written as:

$$\mathcal{L}_{\text{FPA}(\cdot)} = -\mathbb{E}_{(x,y_w,y_l)\sim\mathcal{D}}\left[\text{sg}[C_w(\hat{\pi}_\theta)]\log\pi_\theta(y_w\mid x) - \text{sg}[C_l(\hat{\pi}_\theta)]\log\pi_\theta(y_l\mid x)\right] \tag{6}$$

where $\text{sg}[\cdot]$ denotes the stop-gradient operator. Note that estimating $\hat{\pi}_\theta$ does not require any additional forward passes, as the required $h_\theta$ and $h_{\text{ref}}$ are already obtained, making it computationally efficient (see Appendix C.2).

Table 2: Benchmark results on GSM8K and MATH. FPA shows consistent improvements over the base algorithms DPO, RPO, and SimPER. The difference after applying FPA is displayed in $\Delta$. All results are reported as pass@1 accuracy $\pm$ standard errors from 8 trials with temperature sampling of 0.7. The best performance in each column is bolded, and the second best is underlined.

| Method | Qwen2.5-Math-7B | | DeepSeekMath-7B | | Llama-3.2-3B-Instruct | | Avg. |
|---|---|---|---|---|---|---|---|
| | MATH500 | GSM8K | MATH500 | GSM8K | MATH500 | GSM8K | |
| Base | $63.25_{\pm0.45}$ | $75.10_{\pm0.23}$ | $30.70_{\pm0.56}$ | $56.66_{\pm0.25}$ | $39.00_{\pm0.77}$ | $68.53_{\pm0.16}$ | 55.48 |
| SFT | $65.88_{\pm0.42}$ | $74.40_{\pm0.40}$ | $31.25_{\pm0.40}$ | $61.37_{\pm0.31}$ | $41.58_{\pm0.28}$ | $69.15_{\pm0.16}$ | 57.27 |
| KTO | $\underline{68.15}_{\pm0.40}$ | $76.54_{\pm0.27}$ | $33.88_{\pm0.54}$ | $70.61_{\pm0.23}$ | $\underline{44.27}_{\pm0.64}$ | $74.54_{\pm0.38}$ | 61.33 |
| DPOP | $65.45_{\pm0.55}$ | $79.43_{\pm0.28}$ | $33.73_{\pm0.36}$ | $68.31_{\pm0.26}$ | $43.00_{\pm0.46}$ | $69.86_{\pm0.16}$ | 59.96 |
| DPO | $65.58_{\pm0.38}$ | $80.15_{\pm0.24}$ | $33.20_{\pm0.43}$ | $69.21_{\pm0.27}$ | $43.03_{\pm0.28}$ | $75.28_{\pm0.23}$ | 61.08 |
| +FPA | $66.47_{\pm0.58}$ | $81.96_{\pm0.29}$ | $\underline{34.88}_{\pm0.39}$ | $70.57_{\pm0.21}$ | $44.21_{\pm0.28}$ | $75.71_{\pm0.16}$ | 62.30 |
| $\Delta$ | +0.89 | +1.81 | +1.68 | +1.36 | +1.18 | +0.43 | +1.22 |
| RPO | $65.08_{\pm0.38}$ | $80.34_{\pm0.29}$ | $33.38_{\pm0.32}$ | $\underline{70.87}_{\pm0.24}$ | $43.05_{\pm0.47}$ | $75.00_{\pm0.23}$ | 61.29 |
| +FPA | $68.05_{\pm0.46}$ | $\mathbf{83.80}_{\pm0.23}$ | $33.90_{\pm0.39}$ | $70.84_{\pm0.19}$ | $43.40_{\pm0.41}$ | $\underline{79.27}_{\pm0.18}$ | $\underline{63.21}$ |
| $\Delta$ | +2.97 | +3.46 | +0.52 | −0.03 | +0.35 | +4.27 | +1.92 |
| SimPER | $67.05_{\pm0.45}$ | $79.94_{\pm0.27}$ | $33.92_{\pm0.52}$ | $69.12_{\pm0.25}$ | $42.33_{\pm0.28}$ | $79.03_{\pm0.20}$ | 61.90 |
| +FPA | $\mathbf{72.80}_{\pm0.52}$ | $\underline{82.06}_{\pm0.31}$ | $\mathbf{36.12}_{\pm0.26}$ | $\mathbf{70.89}_{\pm0.35}$ | $\mathbf{45.67}_{\pm0.44}$ | $\mathbf{79.31}_{\pm0.22}$ | $\mathbf{64.48}$ |
| $\Delta$ | +5.75 | +2.12 | +2.20 | +1.77 | +3.34 | +0.28 | +2.58 |

## 4 EXPERIMENTS

### 4.1 EXPERIMENTAL SETUP

We employ three foundation models: two math-specialized models, Qwen2.5-Math-7B (Yang et al., 2024) and DeepSeekMath-7B (Shao et al., 2024), and one general-purpose model, Llama-3.2-3B-Instruct (Grattafiori et al., 2024). We conduct training and evaluation on the widely used MATH (Hendrycks et al., 2021) and GSM8K (Cobbe et al., 2021) benchmarks. We construct each training preference dataset via temperature sampling and regex matching for correctness. A 5% portion of the training dataset is held out for validation. The extrapolation hyperparameter in FPA is set to 0.5 by default after exploration.

We benchmark our approach against several established methods, including Supervised Fine-Tuning (SFT) (Yuan et al., 2023), DPO (Rafailov et al., 2023), DPO-Positive (DPOP) (Pal et al., 2024), Reasoning Preference Optimization (RPO) (Pang et al., 2024), Kahneman-Tversky Optimization (KTO) (Ethayarajh et al., 2024), and SimPER (Xiao et al., 2025). To demonstrate the broad applicability of our regularizer, we apply FPA to three representative algorithms: DPO, a standard baseline with symmetric $C_w = C_l$; RPO, which strengthens preferred gradients for reasoning tasks; and SimPER, which allows asymmetrical control over relative gradient magnitudes. Further details on the experimental settings are available in the Appendix D.

### 4.2 EXPERIMENTAL RESULTS

Our primary findings are summarized in Table 2. FPA demonstrates consistent performance improvements across all tested algorithms, models, and datasets, outperforming the corresponding baselines in nearly every configuration. Notably, applying FPA to SimPER with the Qwen2.5-Math-7B model yields the largest gains (up to 5.75% on MATH and 2.12% on GSM8K), supporting our analysis that algorithms with asymmetric coefficient structures benefit most from FPA's targeted regularization. Importantly, DPO and RPO also show consistent gains. In DPO (symmetric $C_w = C_l$), the improvements reflect safer, symmetrically regularized update steps. By contrast, RPO exhibits a mild asymmetry ($C_w = C_l + \alpha/|y|$), so the dispreferred targeted regularization yields larger gains than DPO. Overall, the ordering of improvements with FPA—DPO < RPO < SimPER—reflects the degree of coefficient asymmetry, as hypothesized in Section 3.

**Training Dynamics of FPA.** To understand its empirical effects, we analyze the training dynamics of FPA. As shown in Figure 3, it consistently maintains higher log probabilities for both preferred

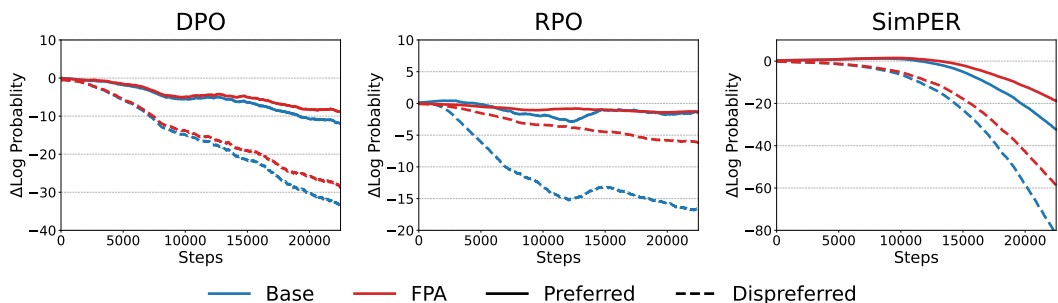

Figure 3: Log-probability difference of $\log \pi_\theta(y|x) - \log \pi_{\text{ref}}(y|x)$ for DPO, RPO, and SimPER before and after applying FPA on the MATH dataset with Qwen2.5-Math-7B.

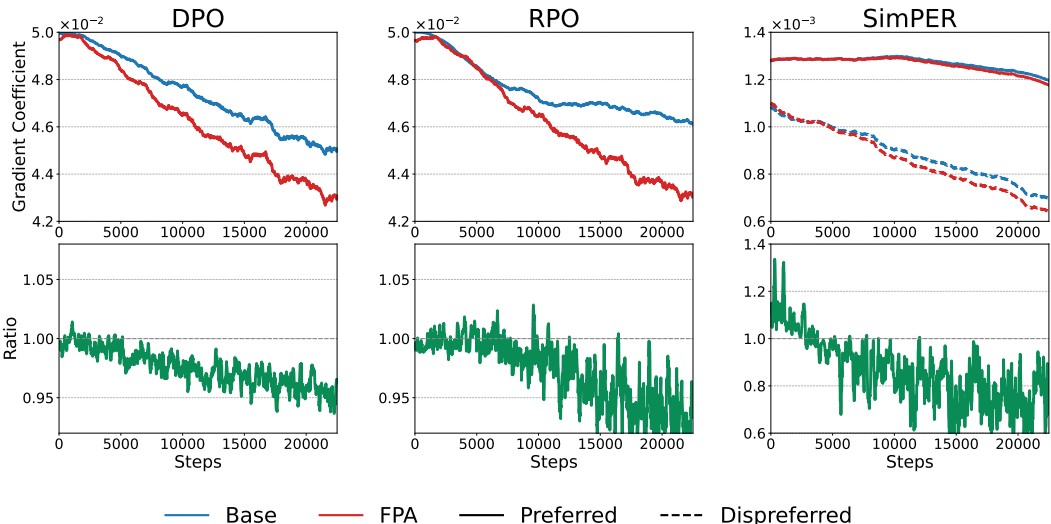

Figure 4: Gradient coefficient $C_l$ for DPO, RPO and both $C_w$ and $C_l$ for SimPER on the MATH dataset with Qwen2.5-Math-7B (top). Note that $C_w = C_l$ for DPO and $C_w = C_l + \alpha/|y|$ in RPO. Ratio between before and after FPA for $C_l$ (bottom).

and dispreferred trajectories throughout training, while still preserving the necessary preference gap between them. This outcome is beneficial, as higher log probabilities indicate that shared useful tokens are not being excessively penalized, which helps prevent the degradation that often arises when both trajectories are pushed to low likelihoods.

Figure 4 shows how this effect arises through the behavior of the coefficient $C$. Lower values of the coefficient $C$ correspond to stronger regularization, which helps preserve the log probabilities of shared tokens. The asymmetric gradient control in SimPER is shown in the right panel, where $C_w$ remains nearly unchanged while $C_l$ is regularized, resulting in a targeted regularization. Figure 4 (bottom) further shows that the regularization begins with no effect (i.e. ratio = 1) and then gradually decreases as training progresses, acting as a proactive brake. Finally, the fluctuations in this ratio highlight its adaptiveness, distinguishing it from a fixed rescaling: it engages only when the predicted future policy signals instability.

## 4.3 FURTHER ANALYSIS

In this section, we conduct further analysis on the effect of FPA. Unless stated otherwise, we use our best-performing model, Qwen2.5-Math-7B, with the SimPER algorithm, as SimPER benefits the most from FPA both theoretically and empirically.

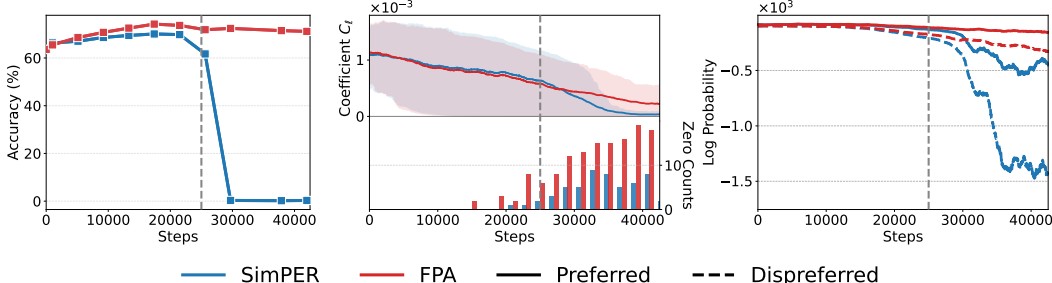

Figure 5: Prolonged training leads to model degradation (dashed line around 25K steps) with Sim-PER on the MATH dataset (left). The coefficient $C_l$ and the number of times it drops to near zero with threshold $10^{-8}$ (center). The log probabilities for preferred and dispreferred trajectories (right).

**Model Degradation.** As shown in Figure 5 (left), while extended training often leads to a performance collapse (dashed line around 25K steps) without FPA, applying FPA successfully prevents this. To analyze this effect more deeply, we plotted the dispreferred gradient coefficient $C_l$ (center-top) and the frequency with which it drops close to zero (center-bottom). Our analysis reveals that FPA's regularization is not just consistently lower; it also adaptively drives the $C_l$ coefficient toward zero more frequently and earlier than SimPER (indicated by Figure 5 center-bottom), particularly just before the onset of model degradation. This demonstrates FPA's proactive nature, as it preemptively nullifies potentially model-degrading updates. In contrast, we can observe the reactive nature of SimPER's regularization, where its $C_l$ coefficient drops significantly, but only well after the performance collapse has already occurred. The log-probability plot (right) shows that this performance collapse correlates with a sharp decrease in the log probabilities for both preferred and dispreferred trajectories. This suggests that low probabilities of dispreferred trajectories do not always indicate a performance gain, but can instead lead to the over-penalization of shared, useful math tokens. It is noteworthy that similar observations on DPO's performance collapse have been reported by Pal et al. (2024).

**Learning Rate vs. FPA.** We first investigate if FPA's regularization differs from simply reducing the gradient size via learning rate. To test this hypothesis, we engineer learning rate schedules for the baseline methods that mimic the decay patterns observed with FPA, further reducing the final rates to 90% for DPO/RPO and 80% for SimPER (Figure 4). However, as shown in Table 3, this manual decay yields no performance benefit. This result indicates that FPA's advantage does not simply come from slower learning, but from its adaptive regularization effect.

**Analysis of $\lambda$.** We explore the relationship between the extrapolation hyperparameter $\lambda$ and the performance gain from FPA. Figure 6 shows that moderate values, such as $\lambda \in [0.5, 1, 2]$, yield consistent and significant gains over the baseline. However, a value an order of magnitude larger, like $\lambda = 10$, results in diminishing returns, with performance similar to the baseline SimPER. This suggests that while proactive regularization is beneficial, excessive extrapolation can be counterproductive. By penalizing the policy based on a future state that is too distant, it can over-regularize the learning signal. The effect of varying $\lambda$ over $C$ is illustrated in Appendix D.6.

Table 3: Performance comparison for lower learning rates.

| Method | MATH500 | GSM8K |
|---|---|---|
| DPO | $65.68_{\pm 0.45}$ | $80.15_{\pm 0.24}$ |
| +Low LR | $66.80_{\pm 0.42}$ | $80.71_{\pm 0.71}$ |
| +FPA | $66.47_{\pm 0.58}$ | $81.96_{\pm 0.29}$ |
| RPO | $65.08_{\pm 0.38}$ | $80.24_{\pm 0.29}$ |
| +Low LR | $66.45_{\pm 0.53}$ | $80.81_{\pm 0.53}$ |
| +FPA | $68.05_{\pm 0.46}$ | $83.80_{\pm 0.23}$ |
| SimPER | $67.05_{\pm 0.45}$ | $79.97_{\pm 0.27}$ |
| +Low LR | $66.92_{\pm 0.34}$ | $79.82_{\pm 0.28}$ |
| +FPA | $72.80_{\pm 0.52}$ | $82.06_{\pm 0.31}$ |

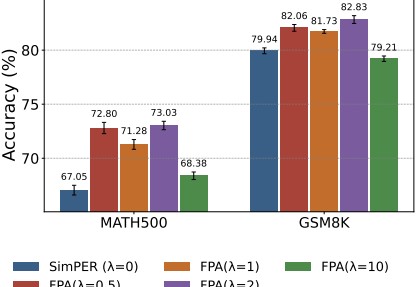

Figure 6: Performance of FPA with different extrapolation strengths $\lambda$.

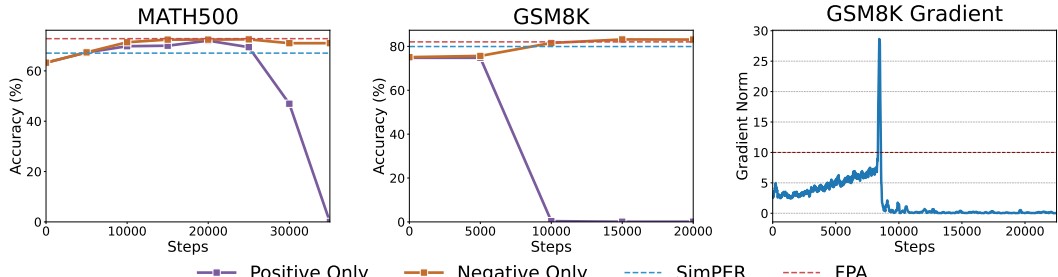

Figure 7: (Left & Center) Performance for targeted FPA, where 'Negative only' and 'Positive only' imply FPA is extrapolating only that component. The performance for SimPER and full FPA from Table 2 is included for reference. (Right) Gradient norm during 'Positive only' training on GSM8K, clipped at 10.

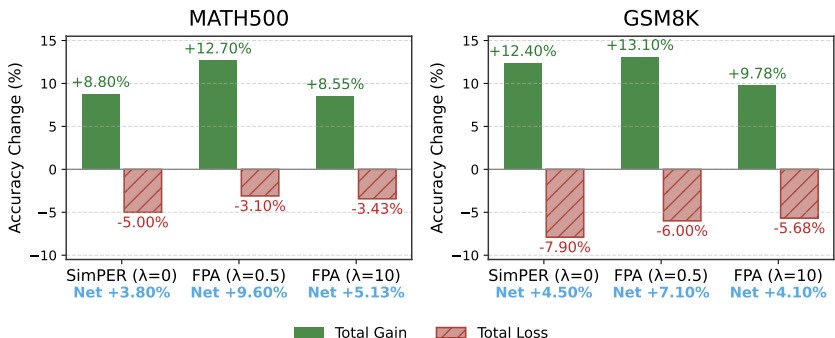

Figure 8: The Total Gain and Total Loss for different extrapolation $\lambda$ values over MATH500 and GSM8K. Note the Net Gain matches the performance gain against base model in Table 2.

**Targeted FPA.** To confirm that regularizing the dispreferred coefficient ($C_l$) is the key factor behind FPA's success, we conduct an ablation study applying its extrapolation to only one coefficient at a time. The results in Figure 7 are clear: applying FPA solely to the dispreferred coefficient is sufficient to prevent model degradation and yields performance comparable to that of the full FPA. In contrast, applying it only to the preferred coefficient ($C_w$) results in a performance collapse similar to the SimPER baseline. This confirms that FPA's key mechanism is its control over the dispreferred gradient, which appears to be the primary source of instability for preference learning in mathematical domains. Additionally, Figure 7 (right) illustrates the actual training gradient norm during performance collapse when we use the 'Positive only' FPA variant. We can see that the gradient norm peaks around model collapse showing the exploding gradient discussed in § 2, then after a while, the norm reduces to 0 after the performance collapse has occurred, showing the reactive nature of SimPER.

**Question-level Analysis.** While Table 2 and Figure 6 show clear improvements, they do not reveal whether FPA succeeds by solving new items or by mitigating forgetting. We therefore perform a question-level decomposition of net accuracy into two components: *Total Gain*, representing newly solved problems, and *Total Loss*, representing previously solved problems that the model now fails (i.e., forgetting). A more detailed explanation of how Total Gain and Total Loss are calculated is provided in Appendix D.5. As shown in Figure 8, while the net performance increases, standard preference learning with SimPER suffers from significant forgetting. In contrast, applying FPA ($\lambda = 0.5$) not only achieves a higher Total Gain but, more importantly, substantially reduces Total Loss, the key factor behind FPA's superior performance.

Additionally, while the highly regularized case ($\lambda = 10$) yields performance comparable to the base SimPER, although its Total Gain is lower, it also exhibits significantly less forgetting, demonstrating a safer training with a lower risk of performance degradation. This highlights the twofold

benefits of FPA: it enhances the learning of new problems while simultaneously preserving existing capabilities, leading to more robust and reliable performance gains.

## 5 RELATED WORKS

### 5.1 IMPROVING REASONING

Reasoning is a well-suited domain for reinforcement learning-based methods due to the relative ease of verifying final answers and the scarcity of high-quality chain-of-thought (CoT) data. One such approach is Reinforcement Learning from Verifiable Rewards (RLVR) (Lambert et al., 2024; Shao et al., 2024; Guo et al., 2025), which directly optimizes models using feedback from verifiers. However, the high computational cost and instability of online RL have motivated more efficient alternatives. For example, the Self-Taught Reasoner (STaR) pipeline (Zelikman et al., 2022) iteratively generates rationales, filters for those that lead to correct answers, and then fine-tunes the model on these successful trajectories. Similarly, ReST-EM (Gulcehre et al., 2023) uses a verifier to collect and fine-tunes on correct examples. While these methods can be understood as offline alternatives to RLVR, they rely solely on supervised fine-tuning (SFT) on correct samples. This limits their learning signal, as they do not incorporate negative gradients from incorrect samples, a key feature of preference learning.

### 5.2 PREFERENCE LEARNING

Direct Preference Optimization (DPO) has shown great success in replacing the unstable and computationally expensive process of Reinforcement Learning from Human Feedback (RLHF) by learning directly from preferred and dispreferred samples. Since the advent of DPO, various algorithms have been proposed to further advance preference learning (Hong et al., 2024; Gheshlaghi Azar et al., 2024; Meng et al., 2024). Notably, KTO (Ethayarajh et al., 2024) demonstrated the ability to train from only desirable/undesirable feedback rather than explicit pairs, and SimPER (Xiao et al., 2025) simplified the alignment process by removing key hyperparameters from the loss function. Despite their success, DPO and other preference learning algorithms often struggle in mathematical and reasoning domains (Pal et al., 2024; Lai et al., 2024) due to gradient entanglement (Yuan et al., 2025). Several methods have been proposed to mitigate this problem. RPO (Pang et al., 2024) adds a negative log-likelihood (NLL) term to explicitly increase the probability of chosen pairs. DPOP (Pal et al., 2024) introduces a term to prevent this probability from decreasing, while Cal-DPO (Xiao et al., 2024) adds a calibration term to simultaneously increase the probability of chosen responses and decrease that of rejected ones. However, all of these algorithms rely on the current policy for regularization, making them fundamentally reactive. In contrast, our solution is proactive, anticipating future policy shifts to prevent instability.

## 6 CONCLUSION

In this paper, we introduce Future Policy Aware (FPA) preference learning, a method that stabilizes preference learning through a proactive future policy estimation. This future policy is estimated via a lightweight logit-space extrapolation from a reference model toward the current model. FPA is a general paradigm that can be applied to any preference learning algorithm that relies on probabilities from the policy model; our experiments show consistent performance improvements when applied to DPO, RPO, and SimPER on the MATH and GSM8K benchmarks. The benefits are most pronounced for algorithms like SimPER that feature asymmetric gradient coefficients, where FPA provides targeted regularization of the unstable dispreferred gradient. Empirical analysis confirms that FPA acts as an adaptive regularizer, offering advantages beyond a simple learning rate reduction. By preemptively suppressing problematic gradients, FPA prevents catastrophic performance collapse during prolonged training and reduces the forgetting of existing capabilities, leading to more stable and robust performance gains. In summary, FPA provides a general and computationally efficient solution to gradient entanglement, applicable to various preference learning algorithms.

## THE USE OF LARGE LANGUAGE MODELS

We used Large Language Models during the preparation of this paper to proofread and improve the readability of the text, to assist in searching for related work, and to provide coding help such as debugging and generating code snippets. The model was not used to generate research ideas, results, or analysis, and all conceptual contributions, experiments, and conclusions are solely those of the authors.

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

# A PILOT EXPERIMENT DETAILS

This section provides the detailed settings for the pilot experiment in Figure 2.

## A.1 DATASETS

To test the gradient entanglement between preferred and dispreferred trajectories, we plot their angles and norms for three different datasets. The difference in trajectory length between the three datasets is illustrated in Figure 9.

- **Stanford IMDb** (Maas et al., 2011): A movie classification dataset with single-token (Positive/Negative) preferred and dispreferred trajectories.
- **Anthropic Helpful and Harmless** (Bai et al., 2022): A commonly used RLHF dataset with moderately long preferred and dispreferred trajectories.
- **Hendrycks MATH dataset** (Hendrycks et al., 2021): A commonly used math dataset with long, gradient entangled preferred and dispreferred trajectories.

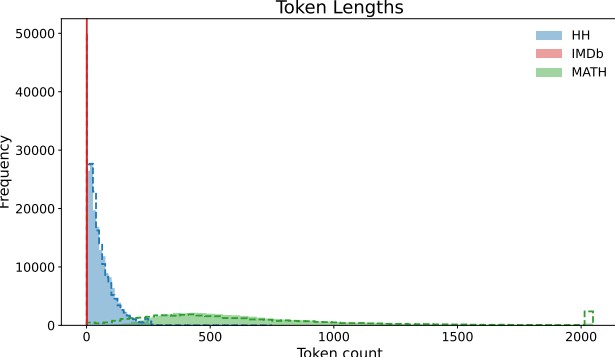

Figure 9: Token lengths for each dataset. IMDb consists of 1 token long trajectories, HH with moderately long trajectories, while MATH consists of long trajectories.

## A.2 TRAINING SETTINGS

We fine-tune the Qwen2.5-1.5B model (Qwen et al., 2024) on the IMDb, HH, and MATH datasets using the SimPER algorithm (Xiao et al., 2025). For the MATH dataset, we use the same preference data as described below for the Qwen2.5-Math-7B model. For IMDb and HH, we use the originally provided datasets. We conduct all training under a single NVIDIA A100 GPU with 80GB of RAM. We use a cosine learning rate scheduler. Table 4 shows the detailed hyperparameters.

Table 4: Pilot experiment hyperparameters.

| Parameter | Value |
| --- | --- |
| Seed | 42 |
| Optimizer | AdamW ($\beta_1 = 0.9, \beta_2 = 0.999$) |
| Warmup Steps | 150 |
| Learning Rate | $5 \times 10^{-6}$ |
| Max Gradient Norm | 10 |
| Max Steps | 25,000 |
| Batch Size | 64 |
| Precision | `bfloat16` |
| Max Sequence Length | 2,048 |

# B    THEORETICAL BACKGROUND

In this section, we theoretically analyze the robustness of DPO and SimPER when applied to mathematical reasoning tasks.

## B.1    LIMITATIONS OF DPO WITH DETERMINISTIC PREFERENCES

Direct Preference Optimization (DPO) may not be robust in mathematical reasoning tasks due to its reliance on the Bradley-Terry model (Bradley & Terry, 1952). Under this model, DPO assumes that preference pairs are selected based on an unknown reward function $r^*(x, y)$, where the probability that response $y_w$ is preferred over $y_l$ follows:

$$p^*(y_w \succ y_l \mid x) = \frac{\exp(r^*(x, y_w))}{\exp(r^*(x, y_w)) + \exp(r^*(x, y_l))} = \sigma\big(r^*(x, y_w) - r^*(x, y_l)\big). \tag{7}$$

This assumption becomes problematic in mathematical reasoning tasks, where preference pairs consist of clearly correct and incorrect answers, yielding deterministic preferences: $p^*(y_w \succ y_l \mid x) = 1$. As Gheshlaghi Azar et al. (2024) and Fisch et al. (2025) show, deterministic preferences require $r^*(y_w) - r^*(y_l) \to \infty$ in the Bradley-Terry model, forcing $\pi_{\theta^*}(y_l \mid x) = 0$ regardless of the KL regularization strength $\beta$. Since mathematical reasoning trajectories often share a large number of common tokens between preferred and dispreferred sequences, this over-penalization of the dispreferred sequences can lead to a degradation of the model's overall performance.

## B.2    THEORETICAL ADVANTAGES OF SIMPER

Unlike DPO's reliance on the Bradley-Terry model, SimPER can be understood as an offline policy gradient method with sequence-length normalization. This formulation aligns naturally with the deterministic reward structure in the mathematical reasoning domain. It makes the learning objective coincide with the data-generating process, thereby eliminating model assumption misspecification.

We begin by establishing the notation used throughout our analysis.

**Notation.**    Let $\mathcal{X}$ be the input space and $\mathcal{Y}^*$ the set of all finite token sequences. For $y \in \mathcal{Y}^*$, let $|y|$ denote its length. We denote the language model policy by $\pi_\theta$, with parameters $\theta \in \Theta$. Given an input $x \in \mathcal{X}$, a response $y \in \mathcal{Y}^*$ is generated from the policy, written as

$$y \sim \pi_\theta(\cdot \mid x).$$

The policy factorizes as

$$\pi_\theta(y \mid x) = \prod_{t=1}^{|y|} \pi_\theta\big(y_t \mid x, y_{<t}\big).$$

For each input $x$, we use a fixed reference policy $\pi_{\text{ref}}$ to generate a pairwise preference set

$$S_x = \{y_w(x), y_l(x)\} \subset \mathcal{Y}^*,$$

where $y_w$ denotes the preferred response and $y_l$ the dispreferred response. We denote the training dataset by

$$\mathcal{D} = \{(x, S_x)\}_{i=1}^N.$$

Throughout, we assume $\pi_\theta(y \mid x) > 0$ for all $y \in S_x$ so that $\log \pi_\theta(y \mid x)$ is well-defined.

To investigate the alignment between SimPER and REINFORCE objectives, we start from the RE-INFORCE objective and define its length-normalized variant. We consider the REINFORCE objective for paired offline preference data with masked rewards $R$ in Definition 1.

**Definition 1.** *Define the masked reward $R : \mathcal{X} \times \mathcal{Y}^* \to \{-1, 0, +1\}$ by*

$$R(x, y) = \begin{cases} +1, & y = y_w(x), \\ -1, & y = y_l(x), \\ 0, & y \notin S_x. \end{cases} \tag{8}$$

Since $R(x, y) = 0$ for $y \notin S_x$, the REINFORCE objective for a fixed input $x$ is

$$J_x(\theta) = \sum_{y \in S_x} \pi_\theta(y \mid x) \, R(x, y). \tag{9}$$

Taking the expectation over the dataset $\mathcal{D}$ yields

$$J(\theta) = \mathbb{E}_{(x, S_x) \sim \mathcal{D}}\big[J_x(\theta)\big]. \tag{10}$$

Based on equation 10, we now define the length-normalized variant.

**Definition 2** (Length-normalized REINFORCE objective). *Motivated by GSPO's sequence-level objective (Zheng et al., 2025)—which clips entire responses to exclude overly off-policy samples and applies length normalization to curb variance and place sequence-level importance ratios on a common scale—we likewise adopt length normalization and define the following REINFORCE variant:*

$$J_{\mathrm{LN}}(\theta) := \mathbb{E}_{(x, S_x) \sim \mathcal{D}}\left[\sum_{y \in S_x} \pi_\theta(y \mid x)^{1/|y|} \, R(x, y)\right]. \tag{11}$$

We can now state the following proposition showing that SimPER implicitly optimizes the length-normalized REINFORCE objective.

**Proposition 1** (SimPER aligns with length-normalized REINFORCE). *The gradient of SimPER's objective equals the gradient of the length-normalized REINFORCE objective as follows:*

$$\nabla_\theta J_{\mathrm{SimPER}}(\theta) = \nabla_\theta J_{\mathrm{LN}}(\theta).$$

*Proof.* We start from SimPER's objective function:

$$J_{\mathrm{SimPER}}(\theta) = -\mathcal{L}_{\mathrm{SimPER}}(\theta)$$

$$= \mathbb{E}_{(x, S_x) \sim \mathcal{D}}\left[\exp\!\left(\tfrac{1}{|y_w|} \log \pi_\theta(y_w \mid x)\right) - \exp\!\left(\tfrac{1}{|y_l|} \log \pi_\theta(y_l \mid x)\right)\right]. \tag{12}$$

Let

$$f_\theta(y \mid x) := \exp\!\left(\tfrac{1}{|y|} \log \pi_\theta(y \mid x)\right).$$

Using this notation, SimPER's objective can be written as:

$$J_{\mathrm{SimPER}}(\theta) = \mathbb{E}_{(x, S_x) \sim \mathcal{D}}[f_\theta(y_w \mid x) - f_\theta(y_l \mid x)].$$

Taking the derivative with respect to $\theta$, we obtain:

$$\nabla_\theta J_{\mathrm{SimPER}}(\theta) = \mathbb{E}_{(x, S_x) \sim \mathcal{D}}[\nabla_\theta f_\theta(y_w \mid x) - \nabla_\theta f_\theta(y_l \mid x)]$$

$$= \mathbb{E}_{(x, S_x) \sim \mathcal{D}}\Big[\pi_\theta(y_w \mid x)^{1/|y_w|} \, |y_w|^{-1} \, \nabla_\theta \log \pi_\theta(y_w \mid x)$$

$$- \pi_\theta(y_l \mid x)^{1/|y_l|} \, |y_l|^{-1} \, \nabla_\theta \log \pi_\theta(y_l \mid x)\Big]. \tag{13}$$

Since $R(x, y_w) = +1$ and $R(x, y_l) = -1$, we can rewrite equation 13 as:

$$\nabla_\theta J_{\mathrm{SimPER}}(\theta) = \mathbb{E}_{(x, S_x) \sim \mathcal{D}}\left[\sum_{y \in S_x} \pi_\theta(y \mid x)^{1/|y|} \, |y|^{-1} \, \nabla_\theta \log \pi_\theta(y \mid x) \, R(x, y)\right]. \tag{14}$$

On the other hand, recall the length-normalized REINFORCE objective from Definition 2:

$$J_{\mathrm{LN}}(\theta) = \mathbb{E}_{(x, S_x) \sim \mathcal{D}}\left[\sum_{y \in S_x} \pi_\theta(y \mid x)^{1/|y|} \, R(x, y)\right]. \tag{15}$$

Differentiating both sides of equation 15 with respect to $\theta$ yields:

$$\nabla_\theta J_{\mathrm{LN}}(\theta) = \mathbb{E}_{(x, S_x) \sim \mathcal{D}}\left[\sum_{y \in S_x} \nabla_\theta \left(\pi_\theta(y \mid x)^{1/|y|}\right) R(x, y)\right]$$

$$= \mathbb{E}_{(x, S_x) \sim \mathcal{D}}\left[\sum_{y \in S_x} \pi_\theta(y \mid x)^{1/|y|} \, |y|^{-1} \, \nabla_\theta \log \pi_\theta(y \mid x) \, R(x, y)\right]. \tag{16}$$

Since the right-hand sides of equation 14 and equation 16 are identical, the following equality holds:

$$\nabla_\theta J_{\mathrm{SimPER}}(\theta) = \nabla_\theta J_{\mathrm{LN}}(\theta), \tag{17}$$

which completes the proof. □

## C  FURTHER DETAILS ON FPA

### C.1  EXACT FORMULATION OF FPA

This section provides the exact equations for applying FPA to DPO, RPO and SimPER using Equations 1,5.

**DPO (with FPA)**

$$\mathcal{L}_{\text{FPA-DPO}} = -\mathbb{E}_{(x,y_w,y_l)\sim\mathcal{D}}\left[\text{sg}\left[\beta\,\sigma\left(\beta\log\frac{\hat{\pi}_\theta(y_l|x)}{\pi_{\text{ref}}(y_l|x)} - \beta\log\frac{\hat{\pi}_\theta(y_w|x)}{\pi_{\text{ref}}(y_w|x)}\right)\right]\log\pi_\theta(y_w|x)\right.$$

$$\left. - \text{sg}\left[\beta\,\sigma\left(\beta\log\frac{\hat{\pi}_\theta(y_l|x)}{\pi_{\text{ref}}(y_l|x)} - \beta\log\frac{\hat{\pi}_\theta(y_w|x)}{\pi_{\text{ref}}(y_w|x)}\right)\right]\log\pi_\theta(y_l|x)\right]$$

**RPO (with FPA)**

$$\mathcal{L}_{\text{FPA-RPO}} = -\mathbb{E}_{(x,y_w,y_l)\sim\mathcal{D}}\left[\text{sg}\left[\beta\,\sigma\left(\beta\log\frac{\hat{\pi}_\theta(y_l|x)}{\pi_{\text{ref}}(y_l|x)} - \beta\log\frac{\hat{\pi}_\theta(y_w|x)}{\pi_{\text{ref}}(y_w|x)}\right] - \frac{\alpha}{|y_w|}\right)\log\pi_\theta(y_w|x)\right.$$

$$\left. - \text{sg}\left[\beta\,\sigma\left(\beta\log\frac{\hat{\pi}_\theta(y_l|x)}{\pi_{\text{ref}}(y_l|x)} - \beta\log\frac{\hat{\pi}_\theta(y_w|x)}{\pi_{\text{ref}}(y_w|x)}\right)\right]\log\pi_\theta(y_l|x)\right]$$

**SimPER (with FPA)**

$$\mathcal{L}_{\text{FPA-SimPER}} = -\mathbb{E}_{(x,y_w,y_l)\sim\mathcal{D}}\left[\text{sg}\left[\frac{1}{|y_w|}\exp\left(\frac{1}{|y_w|}\log\hat{\pi}_\theta(y_w|x)\right)\right]\log\pi_\theta(y_w|x)\right.$$

$$\left. - \text{sg}\left[\frac{1}{|y_l|}\exp\left(\frac{1}{|y_l|}\log\hat{\pi}_\theta(y_l|x)\right)\right]\log\pi_\theta(y_l|x)\right]$$

where sg$[\cdot]$ denotes the stop-gradient operator, and $\hat{\pi}_\theta$ denotes the estimated future policy. It is worth noting that FPA with $\lambda = 0$ does not yield the same loss values themselves, but the same gradient and hence the same training dynamics and results.

### C.2  COMPUTATIONAL COST OF FPA

We analyze the computational overhead of FPA discussed in § 3. For preference-learning algorithms that already rely on a reference model, such as DPO, FPA introduces virtually no additional cost: it requires no extra forward passes and only requires the marginal cost of logit extrapolation, which is negligible. For reference-free algorithms such as SimPER, FPA does require an additional forward pass through the reference model to compute $h_{\text{ref}}$. However, in our pipeline, the dataset is generated with respect to the reference model from the outset, allowing us to cache the corresponding logits during data generation. As a result, the additional cost reduces to a memory overhead rather than extra compute.

# D MAIN EXPERIMENT DETAILS FROM § 4

## D.1 DATASET CONSTRUCTION

To generate a preference dataset, we sample $K = 8$ times for every question in the training sets of MATH and GSM8K. We use regular expression (regex) matching to identify correct answers. Trajectories with correct answers are labeled as preferred while those with incorrect answers are labeled as dispreferred. Any questions where all generated answers are correct or all are incorrect are discarded. Temperature sampling ($T = 0.7$) is used for dataset generation with a constant seed of 42 and 5% of the data is held out for validation. All inferences are performed on a single node with four NVIDIA A100 80GB GPUs with vLLM (Kwon et al., 2023). Table 5 shows the number of preference pairs generated.

Table 5: Preference Dataset Sizes.

| Model | MATH | GSM8K |
|-------|------|-------|
| Qwen2.5-Math-7B | 28,783 | 12,677 |
| DeepSeekMath | 30,889 | 32,471 |
| Llama3.2-3b-Instruct | 44,960 | 24,912 |

## D.2 BASELINE ALGORITHMS

This section details the baseline methods used in our experiment that are not described in § 2.

**Supervised Fine-Tuning (SFT).** Following Yuan et al. (2023) we fine-tune the model using a standard negative log-likelihood loss, exclusively on the preferred trajectories ($y_w$). The loss function is:

$$\mathcal{L}_{\text{SFT}} = -\mathbb{E}_{(x, y_w) \sim \mathcal{D}}[\log \pi_\theta(y_w|x)]$$

**Kahneman-Tversky Optimization (KTO).** A widely used alignment method that does not require paired preference data (Ethayarajh et al., 2024). For a fair comparison, we use the same dataset as other algorithms, treating trajectories as individual preferred and dispreferred examples. The KTO loss is defined as:

$$\mathcal{L}(\pi_\theta, \pi_{\text{ref}}) = \mathbb{E}_{(x,y) \sim \mathcal{D}}[\lambda_y - v(x, y)]$$

where the terms are defined as:

$$r_\theta(x, y) = \log \frac{\pi_\theta(y|x)}{\pi_{\text{ref}}(y|x)}$$

$$z_0(x) = \frac{1}{|\mathcal{D}|} \sum_{(x', y') \in \mathcal{D}} r_\theta(x', y')$$

$$v(x, y) = \begin{cases} \lambda_w \sigma(\beta(r_\theta(x, y) - z_0(x))) & \text{if } y \text{ is preferred} \\ \lambda_l \sigma(\beta(z_0(x) - r_\theta(x, y))) & \text{if } y \text{ is dispreferred} \end{cases}$$

Here, $\lambda_y$ is either $\lambda_w$ or $\lambda_l$ depending on whether response $y$ is preferred/dispreferred, and $z_0(x)$ represents the mean log-likelihood ratio over the dataset.

**DPO-Positive (DPOP).** : A variant of DPO that adds a penalty term to prevent the model from decreasing the likelihood of preferred trajectories, a common issue in reasoning domains (Pal et al., 2024). To avoid confusion with other hyperparameters, we label the DPOP-specific weight as $\lambda_{\text{DPOP}}$. The loss function augments the standard DPO loss:

$$\mathcal{L}_{\text{DPOP}} = -\mathbb{E}_{(x, y_w, y_l) \sim \mathcal{D}} \left[ \log \sigma \left( \beta \left[ \left( \log \frac{\pi_\theta(y_w|x)}{\pi_{\text{ref}}(y_w|x)} - \log \frac{\pi_\theta(y_l|x)}{\pi_{\text{ref}}(y_l|x)} \right) \right. \right. \right.$$

$$\left. \left. \left. - \lambda_{\text{DPOP}} \cdot \max \left( 0, \log \frac{\pi_{\text{ref}}(y_w|x)}{\pi_\theta(y_w|x)} \right) \right] \right) \right]$$

### D.3  TRAINING DETAILS

We conduct all training under a single node with four NVIDIA A100 80GB GPUs. Since the learning rate plays an important role in preference learning, we conduct a comprehensive search. We use a learning rate of $3 \times 10^{-6}$ for DPO and its variants, and $5 \times 10^{-6}$ for KTO and SimPER. The Supervised Fine-Tuning (SFT) phase uses a standard, higher learning rate of $5 \times 10^{-5}$. We employ early stopping based on the performance of the validation set.

For algorithm-specific parameters, we follow the settings from previous work where possible. The DPO hyperparameter $\beta$ is set to 0.1, following (Pang et al., 2024; Jiao et al., 2024). The RPO hyperparameter $\alpha$ is set to 1, as recommended by its authors (Pang et al., 2024). Similarly, the DPOP hyperparameter $\lambda_{\text{DPOP}}$ is set to 50 (Pal et al., 2024). Following the original KTO paper (Ethayarajh et al., 2024), its $\beta$ is also set to 0.1. Additionally, since our dataset contains a balanced number of preferred and dispreferred samples, the KTO loss weights are set to $\lambda_w = \lambda_l = 1$. For FPA, after exploration, we found that the DeepSeekMath-7B benefits from a larger extrapolation $\lambda$, thus we use 5.

Table 6: Main experiment hyperparameters.

| Parameter | Value |
|---|---|
| Seed | 42 |
| Optimizer | AdamW ($\beta_1 = 0.9, \beta_2 = 0.999$) |
| Warmup Steps | 150 |
| Learning Rate | $\{5 \times 10^{-6}, 3 \times 10^{-6}, 1 \times 10^{-6}, 5 \times 10^{-7}\}$ |
| Max Gradient Norm | 10 |
| Max Steps | 25,000 |
| Batch Size | 64 |
| Precision | `bfloat16` |
| Max Sequence Length | 2,048 |

### D.4  EVALUATION PROTOCOL

We evaluate model performance on the basis of Pass@1 accuracy on the respective test sets. All inference is conducted using the vLLM library (Kwon et al., 2023) on a node with four NVIDIA A100 80GB GPUs. For each problem, we generate $n = 8$ independent solutions using temperature sampling ($T = 0.7$). We report the average accuracy over all 8 tries as follows:

$$\text{Pass@1} = \frac{1}{N} \sum_{i=1}^{N} \frac{c_i}{n}$$

where $N$ is the total number of problems, and $c_i$ is the number of correct solutions for problem $i$. To report statistical significance, we compute the standard error of the mean. This is calculated as $\text{SE} = \text{std}/\sqrt{N}$. All results are presented in the format Pass@1 ± SE.

### D.5  DETAILS ON § 4.3 TOTAL GAIN/LOSS

From the evaluation protocol, we can calculate the question-level accuracy (i.e. number of times a question is correct / 8). We then derive the question-level accuracy gain/loss by comparing performance. We calculate Total Gain by averaging all question-level accuracy gains, and the Total Loss by averaging all question-level accuracy losses.

### D.6  EFFECT OF $\lambda$ ON $C$

Figure 10 illustrates the impact of varying $\lambda$ on $C_w$ and $C_l$ for SimPER. While the effect is marginal for $C_w$ and excessively large $\lambda = 10$ actually decreases $C_w$ by a fair margin. This may explain the weak learning behavior exhibited under high $\lambda$ in Figures 6, 8. Additionally, we can observe that the regularization strength on $C_l$ scales accordingly with a larger $\lambda$.

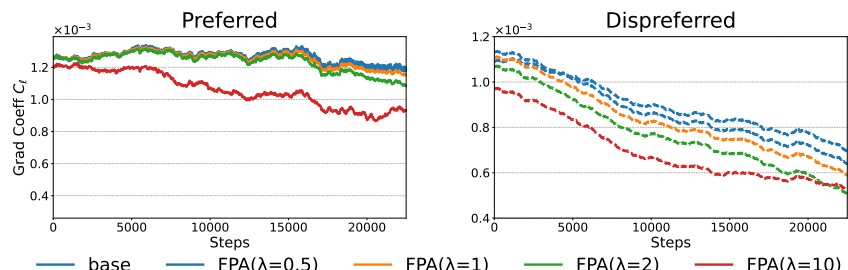

Figure 10: Preferred and dispreferred coefficient sizes for varying $\lambda$ on SimPER

## D.7 PROMPTS

For Qwen2.5-Math-7B and Llama3.2-3b-Instruct, we use the following prompt in Figure 11. For DeepSeekMath-7B, we use few-shot prompting to generate trajectories. (see Figure 12 13)

> Solve this math problem step by step. At the end, make sure to finish the calculation and state the answer exactly once in the following format: The final answer is
> boxed{X}, where X is your final answer.
> Q:{Question}
> A:

Figure 11: Prompt used for Qwen2.5-Math-7B and Llama3.2-3B-Instruct

Problem: Find the domain of the expression $\frac{\sqrt{x-2}}{\sqrt{5-x}}$.

Solution: The expressions inside each square root must be non-negative.
Therefore, $x - 2 \geq 0$, so $x \geq 2$, and $5 - x \geq 0$, so $x \leq 5$.
Also, the denominator cannot be equal to zero, so $5 - x > 0$, which gives $x < 5$.
Therefore, the domain of the expression is $\boxed{[2, 5)}$.
Final Answer: The final answer is $[2, 5)$. I hope it is correct.

Problem: If $\det \mathbf{A} = 2$ and $\det \mathbf{B} = 12$, then find $\det(\mathbf{AB})$. Solution: We have that $\det(\mathbf{AB}) = (\det \mathbf{A})(\det \mathbf{B}) = (2)(12) = \boxed{24}$.
Final Answer: The final answer is 24. I hope it is correct.

Problem: Terrell usually lifts two 20-pound weights 12 times. If he uses two 15-pound weights instead, how many times must Terrell lift them in order to lift the same total weight?
Solution: If Terrell lifts two 20-pound weights 12 times, he lifts a total of $2 \cdot 12 \cdot 20 = 480$ pounds of weight. If he lifts two 15-pound weights instead for $n$ times, he will lift a total of $2 \cdot 15 \cdot n = 30n$ pounds of weight. Equating this to 480 pounds, we can solve for $n$:

$$30n = 480$$
$$\Rightarrow \quad n = 480/30 = \boxed{16}$$

Final Answer: The final answer is 16. I hope it is correct.

Problem: If the system of equations

$$6x - 4y = a,$$
$$6y - 9x = b.$$

has a solution $(x, y)$ where $x$ and $y$ are both nonzero, find $\frac{a}{b}$, assuming $b$ is nonzero.
Solution: If we multiply the first equation by $-\frac{3}{2}$, we obtain

$$6y - 9x = -\frac{3}{2}a.$$

Since we also know that $6y - 9x = b$, we have

$$-\frac{3}{2}a = b \Rightarrow \frac{a}{b} = \boxed{-\frac{2}{3}}.$$

Final Answer: The final answer is $-\frac{2}{3}$. I hope it is correct.

Problem: {Problem}
Solution:

Figure 12: 4-shot Prompt used for MATH in DeepSeekMath-7B(Lewkowycz et al., 2022)

Problem:
Shawn has five toys. For Christmas, he got two toys each from his mom and dad. How many toys does he have now?
Solution:
Shawn started with 5 toys. He received 2 toys from his mom and 2 toys from his dad, for a total of $2 + 2 = 4$ additional toys. Therefore, he now has $5 + 4 = \boxed{9}$ toys.

Problem:
If there are 3 cars in the parking lot and 2 more cars arrive, how many cars are in the parking lot?
Solution:
There are originally 3 cars in the parking lot. When 2 more cars arrive, the total number of cars becomes $3 + 2 = \boxed{5}$.

Problem:
Jason had 20 lollipops. He gave Denny some lollipops. Now Jason has 12 lollipops. How many lollipops did Jason give to Denny?
Solution:
Jason started with 20 lollipops and ended with 12 lollipops. The number of lollipops he gave to Denny is $20 - 12 = \boxed{8}$.

Problem:
There were nine computers in the server room. Five more computers were installed each day, from Monday to Thursday. How many computers are now in the server room?
Solution:
There were originally 9 computers. From Monday to Thursday is 4 days, and 5 computers were installed each day. The total number of computers installed is $5 \times 4 = 20$. Therefore, the total number of computers is $9 + 20 = \boxed{29}$.

Problem:
Leah had 32 chocolates and her sister had 42. If they ate 35, how many pieces do they have left in total?
Solution:
Leah had 32 chocolates and her sister had 42 chocolates. Together they had $32 + 42 = 74$ chocolates. After eating 35 chocolates, they have $74 - 35 = \boxed{39}$ chocolates left.

Problem:
Michael had 58 golf balls. On Tuesday, he lost 23 golf balls. On Wednesday, he lost 2 more. How many golf balls did he have at the end of Wednesday?
Solution:
Michael started with 58 golf balls. After losing 23 on Tuesday, he had $58 - 23 = 35$ golf balls. After losing 2 more on Wednesday, he had $35 - 2 = \boxed{33}$ golf balls.

Problem:
There are 15 trees in the grove. Grove workers will plant trees in the grove today. After they are done, there will be 21 trees. How many trees did the grove workers plant today?
Solution:
There are 15 trees originally. After planting, there are 21 trees total. The number of trees planted is $21 - 15 = \boxed{6}$.

Problem:
Olivia has 23. She bought five bagels for 3 each. How much money does she have left?
Solution:
Olivia had 23. Each bagel costs 3, so 5 bagels cost $5 \times 3 = 15$ dollars. The amount of money she has left is $23 - 15 = \boxed{8}$ dollars.

Problem:{Problem}
Solution:

Figure 13: 8-shot Prompt used for GSM8K in DeepSeekMath-7B(Wei et al., 2022)

