# OpenReview forum: "Future Policy Aware Preference Learning for Mathematical Reasoning"
_ICLR.cc/2026/Conference — ICLR 2026 Conference Withdrawn Submission_

### Official Review · Reviewer_yZuy · 2025-10-30

**Soundness:** 3
**Presentation:** 3
**Contribution:** 3
**Rating:** 6
**Confidence:** 4

**Summary:**

The paper studies why standard preference-learning methods degrade on mathematical reasoning due to gradient entanglement created by token overlap between preferred and rejected trajectories. It argues existing regularizers are reactive and kick in too late, over-penalizing shared useful tokens. The method introduces Future Policy Aware (FPA) preference learning: replace the current-policy probabilities used in the regularization term with a predicted future policy obtained by lightweight logit-space extrapolation from a reference toward the current model, applied via stop-gradient. Experiments apply FPA to DPO, RPO, and SimPER across GSM8K and MATH with several backbones, showing consistent gains, better stability, and negligible overhead, plus analyses of coefficients, angles, and forgetting.

**Strengths:**

1. The paper proposes a simple, general method: a drop-in, reference-aware extrapolation that proactively regularizes gradients without extra forward passes; compatible with multiple preference objectives.
2. The paper also provides a thorough empirical analysis: gradient angle/norm diagnostics, coefficient behavior, targeted ablations (positive-only/negative-only), $\lambda$ sensitivity, and learning-rate controls that isolate the source of benefit beyond simple step-size changes.

**Weaknesses:**

1. The main concern is that the paper emphasizes preference learning from the title through the body, yet does not evaluate on standard preference-learning tasks, i.e., RLHF-related benchmarks such as AlpacaEval 2 [1] and MT-Bench [2]. The work focuses on math reasoning datasets, but on-policy RL methods like RLVR currently perform better than off-policy RL methods on math reasoning (RLVR is at least the mainstream approach). This raises doubts about the necessity of using DPO for math reasoning. If the authors report FPA on RLHF-related tasks, the case would be much stronger.
2. The main motivation is: “Regularization in preference learning currently scales gradients using a coefficient computed under the current policy, which is reactive and often engages only after shared, useful tokens have already been over-penalized.” It is unclear whether there is supporting experimental evidence or a theoretical derivation for this statement.
3. If the focus is solely math reasoning, the experimental tables should add results using the same training data with RLVR. This is the mainstream approach for LLM math reasoning. Readers will be curious about the comparison between FPA and RLVR.
4. In lines 1059–1060, the authors state that the evaluation metric is pass@1 averaged over 8 runs. Given that scores for math reasoning models often vary substantially across runs [3], it is recommended to increase the number of runs.

[1] AlpacaEval: An automatic evaluator of instruction-following models.
[2] Judging LLM-as-a-judge with MT-Bench and Chatbot Arena.
[3] A Sober Look at Progress in Language Model Reasoning: Pitfalls and Paths to Reproducibility

**Questions:**

Please refer to my weakness

---

### Official Review · Reviewer_Vjxt · 2025-10-31

**Soundness:** 2
**Presentation:** 3
**Contribution:** 2
**Rating:** 2
**Confidence:** 4

**Summary:**

The authors aim to improve preference alignment of LLMs on mathematical reasoning tasks, by proposing a new algorithm named Future Policy Aware (FPA) preference learning. The authors pose that current existing regularization techniques are reactive (activates after the model has already started to degrade), while FPA is proactive by estimating a future policy. The future policy is estimated via logit space extrapolation from a reference model, and acts to stabilize preference alignment. This simulates a type of "early brake" method, and experiments demonstrate best improvement when combined with other algorithms on classical mathematical reasoning benchmarks.

**Strengths:**

- Clear and well-written paper that discusses the background of well-known preference alignment algorithms and known challenges. There is discussion of gradient entanglement, and Fig 2 shows that the dispreferred gradient norm is consistently larger than the preferred one (validating this as the source of instability).

- Addresses catastrophic forgetting, which is indeed a problem in preference learning today. Preference learning has recently become ubiquitous in the fine-tuning pipeline for many applications, hence the topic of this paper is well-posed and relevant. Fig 5 in particular shows the catastrophic performance collapse of SimPER after 25,000 steps. Fig 7 summarizes ablation studies well.

- The method is computationally cheap, making it simple and lightweight. Since FPA is a linear extrapolation, it adds negligible computational overhead. The method itself is a simple proactive regularizer, and the authors have identified the practical use-case of mathematical reasoning to demonstrate its efficacy.

**Weaknesses:**

- The strongest results are derived from a combination of FPA and SimPER. However this critically negates the key advantage of SimPER: it's reference-free. FPA introduces the need for a reference model. The authors claim this is reasonable since logits can be cached during data generation, but this ultimately adds data complexity and negates SimPER's main selling point.


- The proposed method is somewhat incremental. FPA is not a new algorithm for preference alignment, but is an additional step that can be added in the preference alignment pipeline. As the community pushes towards scaling large models feasibly, making the most use of compute resources, extracting maximum value from limited data: other stand-alone preference alignment algorithms (DPO, SimPO, ORPO) advertises being scalable and simple (reference-free). This is a somewhat opposite direction from what FPA introduces. Unless the addition of the extra FPA step (incurring reference models, additional hyperparameters, memory) yields dramatic improvements in general preference alignment.


- The paper focuses on the topic of mathematical reasoning as its main motivation, and the gradient entanglement problem is the most severe due to high token overlap. It is unclear if FPA can provide any benefit in other domains, where a dispreferred trajectory might have very little token overlap with a preferred one.


- The proposed future policy is heuristics driven. The mechanism is a first-order linear extrapolation in logit space, and not a model-based prediction of a future state. It is unclear if this extrapolation can capture the non-linear dynamics of the optimizer's future state. Since there is some lack of mathematical interpretability, a wider domain of experiments is likely needed to evaluate this heuristic. Especially since it introduces the additional hyperparameter $\lambda$.

**Questions:**

- Fig 7 shows that adding FPA only to the preferred gradient is not neutral, but causes the model to collapse just like the baseline. This is counter-intuitive. Why does proactively regularizing a "good" gradient result in failure?

- How does FPA interact with the learning rate? With reference to Table 3: if using FPA's proactive regularization should you use a higher/lower learning rate than normal, and is it possible to perform ablation on this effect? Currently the paper uses the same learning rate for baselines and FPA variants, which might be sub-optimal.

- Have the authors experimented with other wide domains in preference learning? Such as summarization, or other tasks. Since the future in FPA is not tied to a specific time step, it's unclear how this heuristic future interacts with actual future states being tracked by the optimizer (i.e. variance estimates, etc). Further experimentation in wider domains might yield more transparency.

---

### Official Review · Reviewer_qnp4 · 2025-11-01

**Soundness:** 3
**Presentation:** 3
**Contribution:** 2
**Rating:** 4
**Confidence:** 3

**Summary:**

This paper introduces "Future Policy Aware" (FPA) preference learning, a general approach to stabilize and improve preference optimization for large language models (LLMs) in the context of mathematical reasoning. Standard preference learning methods like Direct Preference Optimization (DPO) often degrade performance in mathematical domains due to gradient entanglement between preferred and dispreferred trajectories, which share numerous tokens. FPA addresses this by replacing regularization based on the current policy with an anticipatory version based on an extrapolated "future" policy estimated using logit-space interpolation. The approach is shown to provide proactive, adaptive regularization, leading to more stable and effective preference learning. The method is evaluated empirically on MATH and GSM8K benchmarks using three algorithms (DPO, RPO, SimPER) across multiple math-oriented and general LLMs, with FPA delivering consistent improvements.

**Strengths:**

1. Clear Problem Motivation & Importance: The paper clearly articulates the problem of gradient entanglement—an under-addressed yet critical failure mode in mathematical preference learning for LLMs.
2. Mathematical Detail and Theoretical Analysis: The formulation and justification for FPA are well-explained. The derivation of the extrapolated policy (Equation in section 2 and section 3) ensure the computational efficiency and soundness of the approach. Appendix B offers a careful mathematical rationale for why classical DPO may fail on deterministic preference data (as in math reasoning).
3. Low Computational Overhead: Section C.2 states and justifies that FPA incurs negligible or only marginal extra cost, leveraging already computed model logits.

**Weaknesses:**

1. No Exploration of Step-wise Preference Learning: While gradient entanglement is clearly diagnosed as a source of performance degradation, the work restricts itself to trajectory-level preference optimization. Recent research indicates that step-level feedback and optimization can more precisely target undesirable intermediate states, which is especially relevant in mathematical reasoning tasks with compositional structure. The absence of any empirical exploration or theoretical rationale for not considering step-level preference mechanisms leaves an opportunity unexplored
2. Degree of Generalization Claimed Without Evidence: While the paper argues that FPA is broadly applicable to "any preference learning algorithm relying on policy probabilities," the empirical evaluation is limited to three post-training alignment algorithms and two datasets. There is little evidence to support the generalization of FPA to, for example, RLHF variants or reinforcement learning outside mathematical tasks.

References
- Chen, Guoxin, et al. "Step-level value preference optimization for mathematical reasoning." arXiv preprint arXiv:2406.10858 (2024).
- Lai, Xin, et al. "Step-dpo: Step-wise preference optimization for long-chain reasoning of llms." arXiv preprint arXiv:2406.18629 (2024).

**Questions:**

1. How does FPA interact with step-level preference learning: Have you compared FPA with recent advances in step-level or sub-trajectory-based optimization methods (e.g., SVPO, Step-DPO), which have also shown strong performance on mathematical reasoning tasks? Furthermore, can FPA be applied to both trajectory-level and step-level optimization, or are there any technical obstacles that would make such integration difficult?

2. Generalization of FPA: How does FPA perform on LLM tasks that are characterized by low gradient entanglement—for example, generative tasks, open-domain dialogue, or summarization? Could you provide empirical evidence or at least qualitative analysis to confirm that FPA does not degrade performance in such settings?

---

### Official Review · Reviewer_TG7n · 2025-11-04

**Soundness:** 2
**Presentation:** 2
**Contribution:** 1
**Rating:** 2
**Confidence:** 5

**Summary:**

This paper proposes Future Policy Aware (FPA) preference learning, as a plug-in weighting strategy of the preferred and dispreferred samples in DPO-style algorithms.

**Strengths:**

This appears to be a fairly standard DPO-style paper: it introduces a modified loss function, validates results using ≤7B models on math datasets, and conducts typical analyses on metrics and hyperparameters, with an effort to provide more extensive experimental results.

**Weaknesses:**

I find the paper’s motivation, logic, and claimed novelty unconvincing.

1. The motivation for using “regularization” throughout the paper is unclear. For example, in the discussion following Equation (1), the coefficient labeled as a “regularization term” appears mischaracterized; it simply controls the relative strength of components rather than serving as a true regularizer.

2. The relationship between Equation (1) and other loss formulations should be clarified. It would help to derive the gradient of each variant to explicitly show the linear combination between desired and undesired output logits, along with the corresponding coefficients.

3. Equation (5) depends on the current policy $\pi_\theta$, the reference policy $\pi_{\text{ref}}$, and a hyperparameter $\lambda$. The authors should provide its explicit form in terms of $\pi_\theta$ and $\pi_{\text{ref}}$, and explain precisely how it differs from the algorithms in Table 1 that use the hyperparameter $\beta$. A detailed mathematical derivation is needed here.

4. There is no clear justification for why extrapolating the coefficient term would improve performance over the original formulation, especially given the mathematical ambiguity noted in point 3. I also don't see empirical analysis of the coefficient itself, which should be the main focus of this paper.

5. The novelty appears limited with just slight modification of the DPO-style loss functions.

6. The theoretical analysis is unclear. Points 2 and 3 require mathematical derivations, but none are provided. Moreover, Appendix B.2 focuses entirely on SimPER, which appears unrelated to the main contributions of this work.

**Questions:**

Why FPA is better? I cannot see any reason and benefit by replacing a term in the coefficient to the extrapolation.

---

### Note · Authors · 2025-12-01

I have read and agree with the venue's withdrawal policy on behalf of myself and my co-authors.